# Contextual cues are not unique for motor learning: Task-dependant switching of feedback controllers

**Justinas Česonis**[1], **David W. Franklin**[1,2,3]*

**1** Neuromuscular Diagnostics, Department of Sport and Health Sciences, Technical University of Munich, Munich, Germany, **2** Munich Institute of Robotics and Machine Intelligence (MIRMI), Technical University of Munich, Munich, Germany, **3** Munich Data Science Institute (MDSI), Technical University of Munich, Munich, Germany

* david.franklin@tum.de

**Citation:** Česonis J, Franklin DW (2022) Contextual cues are not unique for motor learning: Task-dependant switching of feedback controllers. PLoS Comput Biol 18(6): e1010192. https://doi.org/10.1371/journal.pcbi.1010192

**Data Availability Statement:** All the Jupyter notebooks for the data analysis, pre-processed experimental data and statistical analysis conducted in this article are available at https://doi.org/10.6084/m9.figshare.17113904.v1.

## Abstract

The separation of distinct motor memories by contextual cues is a well known and well studied phenomenon of feedforward human motor control. However, there is no clear evidence of such context-induced separation in feedback control. Here we test both experimentally and computationally if context-dependent switching of feedback controllers is possible in the human motor system. Specifically, we probe visuomotor feedback responses of our human participants in two different tasks—stop and hit—and under two different schedules. The first, blocked schedule, is used to measure the behaviour of stop and hit controllers in isolation, showing that it can only be described by two independent controllers with two different sets of control gains. The second, mixed schedule, is then used to compare how such behaviour evolves when participants regularly switch from one task to the other. Our results support our hypothesis that there is contextual switching of feedback controllers, further extending the accumulating evidence of shared features between feedforward and feedback control.

## Author summary

Extensive evidence has demonstrated that humans can learn distinct motor memories (i.e. independent feedforward controllers) using contextual cues. However, there is little evidence that such contextual cues produce similar separation of feedback controllers. As accumulating evidence highlights the connection between feedforward and feedback control, we propose that context may be used to separate feedback controllers as well. It has not been trivial to test experimentally whether a change in context also modulates the feedback control, as the controller output is affected by other non-contextual factors such as movement kinematics, time-to-target or the properties of the perturbation used to probe the control. Here we present a computational approach based on normative modelling where we separate the effects of the context from other non-contextual effects on the visuomotor feedback system. We then show experimentally that task context

**Funding:** The author received no specific funding for this work.

**Competing interests:** The authors have declared that no competing interests exist.

independently modulates the feedback control in a particular manner that can be reliably predicted using optimal feedback control.

## Introduction

Whether it is touching a hot surface, returning a tennis serve or simply lifting an object, the human body utilises a variety of sensory inputs to produce movements of any complexity. Indeed, different feedback modalities of human motor control, such as stretch reflex [1–3], vestibulo-ocular reflex [4, 5], visuomotor [6–14], or even auditory feedback [15, 16] have extensively been studied in prior literature. However, most studies have investigated feedback control in paradigms of either a single task [17–21], or multiple tasks presented in their own dedicated blocks [22–26]. While such designs provide key insights into the behaviour of the feedback controller in isolation, they are not entirely reflective of human behaviour in real-life situations. For example, a realistic sequence of events could require a volleyball player to first pick up the ball from the ground by reaching for it with their hand and stopping on contact, only then to hit the same ball with the same hand a few moments later while serving. While studying both components independently has received focus in the field of motor control, any interactions between the feedback controllers in the context of rapid switching have not been broadly studied.

While feedback control in human movement is critical in correcting for random errors within movements, feedforward control corrects for movement errors that are predictable. In order to systematically predict and compensate for specific errors upcoming in a given movement, the mechanism of contextual switching via contextual cues is broadly accepted. It is now well understood that performing two opposing tasks in an alternating manner will lead to interference [27–29], resulting in behaviour that is averaged between the two tasks, failing to deal with either task. However, if the two tasks are performed in sufficiently different contexts, such as separate physical or visual workspaces [30–32], or different lead-in [33, 34] or follow-through movements [35, 36], this interference can be reduced, allowing the formation of two separate motor memories. Hence, it is reasonable to expect, that a similar contextual regulation could be present in feedback controllers. Therefore, in this study we test whether the feedback control policies exhibit such modulation when humans are presented with different tasks in an alternating manner.

One difference between studying contextual switching in feedforward and feedback control is that it is difficult to evaluate whether the feedback control policy has changed after the intervention. Specifically, it has been shown computationally that the optimal feedback controller (OFC) with fixed parameters can produce variable responses when correcting for perturbations within the movement, for example, when the comparable perturbations are induced in different parts (e.g. early or late) of otherwise identical movements [26, 37–39]. Furthermore, such behaviour was also observed in experimental studies [7, 20, 26, 39–41]. Hence, merely observing a difference in the feedback response is not enough to conclude a change in the control policy. However, recently we demonstrated that as long as two perturbations of the same magnitude are induced at the same time-to-target (which is defined as a difference between the perturbation onset and movement end), the same feedback control policy produces the same magnitude response, independent of whether the two perturbations occurred at the same location, time from the beginning of the movement, or the movement velocity [26]. Thus, we can utilise this relationship between the magnitude (or intensity) of the feedback

response to a perturbation at the same time-to-target to quantify whether the difference in the response is due to the change in the control policy or not.

There are several studies that have already looked into contextual regulation in feedback control tasks. Most of such studies, to our knowledge, approached this question by modulating the structure of a target (wide vs. narrow, long vs short, etc.) [42–45], or by including obstacles along the reaching path [42, 45]. Results of these studies are consistent with optimal control-like behaviour with separate controllers for different tasks, even when the target structure is changed on random trials [42, 44, 45], however one study suggests that control when the target is unpredictable may be sub-optimal [43]. In this study we test whether such rapid switching also holds true for tasks, where target structure remains unchanged, but the tasks themselves require a change of movement properties. Specifically, we test how the feedback control policies are affected when our participants are presented with a "multitasking" scenario where they have to switch between performing two distinct tasks, i.e. reaching to and stopping at the target, or hitting through the target and stopping behind it. While the two tasks are fundamentally different, and in isolation should require different feedback control policies, here we also test whether the same relationship holds true in the mixed schedule (as it would for contextual switching in feedforward control), or if the interference between two control policies results in a single policy, averaged or weighted between the two independent controllers.

## Results

In this study we tested the behaviour of the human feedback controller when switching between two different tasks. Specifically, we presented our human participants with two tasks requiring different control policies—a stop task, where participants had to reach and stop at the target, and a hitting task, where participants had to punch through the target and stop behind it. In our previous work we demonstrated computationally that these two different types of movements trigger feedback responses of different magnitudes, even if the perturbations occur at the same position, time, or time-to-target [26]. However, if the two movements share the same goal (for example the goal of stopping at the same target), then these feedback responses match in magnitude if the time-to-target matches in both movements, irrespective of other movement parameters like peak velocity, movement distance, distance to the target or current velocity. Therefore, such a relation between time-to-target and feedback response intensity could be used to characterise the feedback control policy.

We use the relationship between the time-to-target and the feedback response intensity (which serves as a proxy for feedback controller gain) as a means to analyse the controller behaviour when the task changes. Specifically, we propose two alternatives for the architecture of such control: a single universal feedback controller that exhibits adaptation to a given task (Fig 1A), or multiple task-specific controllers, gated by task context (Fig 1B). When presented with a single task in a blocked schedule (e.g. blocked stop or blocked hit), both the universal controller and task specific controllers are expected to behave similarly, as the universal controller should easily adapt its gains appropriately for the required task. However, if multiple tasks are presented in a mixed schedule (i.e. task can randomly switch from trial to trial), the different control architectures predict different responses. Particularly, a single universal controller would aim to adapt to each presented task, thus on average producing responses somewhere in between the two given tasks within the mixed schedule (Fig 1C). In contrast, a set of task-specific controllers would produce similar responses in the mixed schedule as they would in a blocked schedule, as for every trial an appropriate controller is selected from a set of controllers, rather than being adapted for the task (Fig 1D).

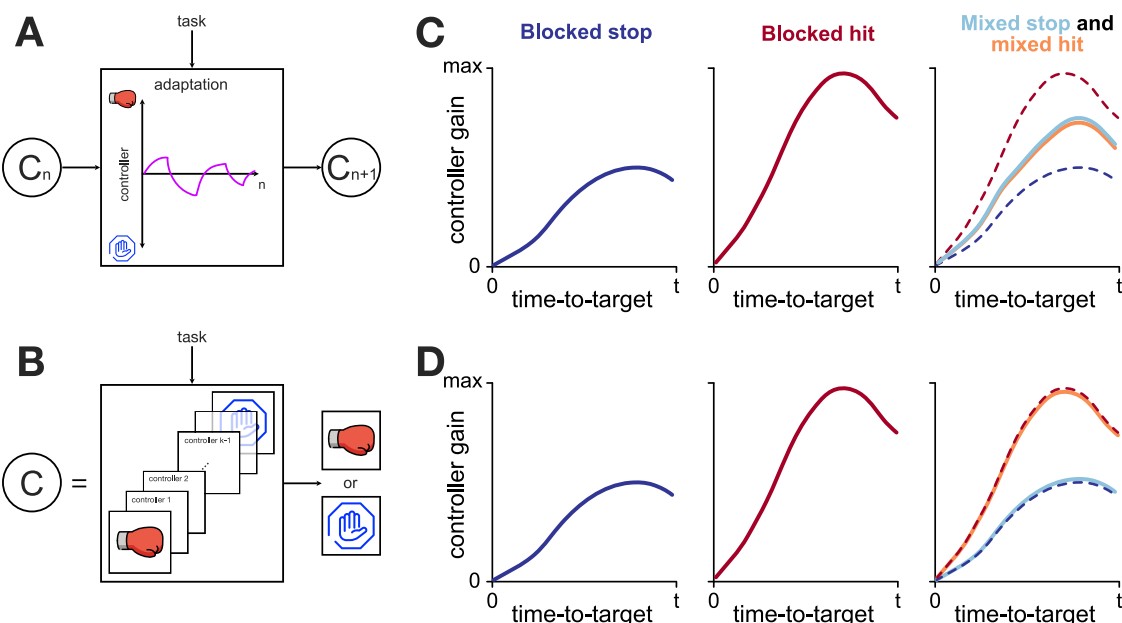

**Fig 1. Theoretical predictions of two different architectures for feedback regulation. A.** Universal feedback controller. A single feedback controller is used to produce both stop and hit movements, and is adapted to the given task over multiple trials. Such adaptive behaviour is reminiscent of the behaviour of the feedforward controller when learning two opposing force-fields without separable context. $C_n$ indicates a feedback controller at trial $n$ **B.** A feedback controller as a set of task-specific controllers. A task-specific controller (stop or hit) is selected based on the task-related context and is used during the given movement. Such contextual switching behaviour is reminiscent of the behaviour of the feedforward controller when learning two opposing force fields with separable context. **C.** Expected regulation of feedback responses by the universal feedback controller. When exposed to a single task for a long time (blocked schedule) the controller adapts to the given task, producing optimal responses for both stop and hit conditions. However, due to interference within the mixed schedule, such a controller would settle to the average (or weighted average) gains between the two blocked conditions. **D.** Expected regulation of feedback responses by a set of task specific controllers. Within the blocked schedule, similar regulation is expected between hit and stop as in the case of the universal controller (**C**). However, in the mixed schedule, due to the task-related context, an appropriate controller is recalled on a trial-to-trial basis, producing similar regulation as within the blocked schedule.

In order to probe the control policies of human participants within these different tasks, we occasionally perturbed participants during the movement by visually shifting the target perpendicular to movement direction and inducing a reactive visuomotor feedback response (Fig 2A). Recently it has become common practice to maintain these perturbations until the end of the movement, such that an active correction is required to successfully complete the trial [19, 26, 41, 46–51]. However, we have noticed in our previous work that such maintained perturbations significantly impact the overall time-to-target, which in turn affects the visuomotor feedback gains [26]. Thus, to keep the measurements of visuomotor feedback responses consistent within time-to-target, in this study we only perturbed our participants laterally in channel trials [7, 40, 52] and maintained these perturbations for 250 ms before switching them off, making any corrections redundant. As a result, even when producing the feedback response, participants' hands are constrained along the path of forward movement, resulting in matching movement durations independent of different perturbation onsets.

Participants produced involuntary feedback responses to the target jumps. These responses, observed as a lateral force exerted by the participants on the handle of the robotic manipulandum, were modulated by the different perturbation onsets (Fig 2B and 2C). From these force responses we computed feedback intensities, by averaging individual responses over a time window 180 ms–230 ms relative to the perturbation onset on each individual trial. This time

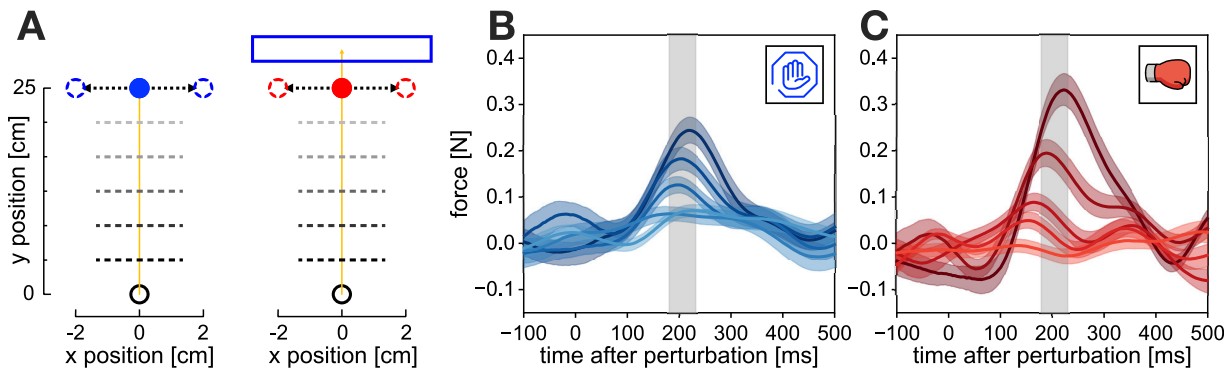

**Fig 2. Experimental perturbations and responses. A**. Perturbations in stop (left) and hit (right) conditions. Participants performed a forward reaching movement towards a target, positioned 25.0 cm in front of the start position. When the hand crossed one of five evenly spaced locations (dashed lines), a perturbation could be induced by shifting the target by 2 cm laterally for 250 ms and then returned back to the original position. Participants were instructed to either stop at the blue target (stop condition), or hit the red target and stop within the blue rectangle (hit condition). **B**. Net unscaled feedback responses to the target perturbations in the stop condition, measured via the force channel. Participants produced corrective responses to the target perturbations that varied by different perturbation onsets. Different traces represent different perturbation onsets, with darker colours indicating earlier perturbations. Shaded areas represent one standard error of the mean (SEM). The grey rectangle represents the time window of 180–230 ms, where the visuomotor feedback intensities are measured. **C**. Net unscaled feedback responses to the target perturbations in the hit condition.

window has now been used in numerous studies to quantify such responses and is associated with the involuntary, early visuomotor responses [7, 25, 40, 50, 51].

## OFC model predicts differences between hit and stop conditions

We utilised the mixed-horizon OFC [38] model, presented in our earlier work, to generate predictions of feedback control policies in our current study. Due to the experimental design of this study not requiring an extension in movement times after perturbations, the predictions of the mixed-horizon model also matched the predictions of our earlier time-to-target OFC model [26]. In order to compare differences in control throughout hit and stop movements, we first simulated two movement conditions: a 25 cm long movement with 60 cm/s peak velocity and velocity at the target distance <1 cm/s (stop condition), and a similar movement, but with velocity at the target >20 cm/s (hit condition) (Fig 3A). Both models were implemented using a linear quadratic regulator (LQR), and were identical, apart from the difference in state-dependent costs of terminal velocity and terminal force. Here we reduced these cost parameters for the hit model by a factor of 50 in order to reduce the incentive to stop at the target, and thus successfully simulate hit-like movements. In addition, we also simulated a third condition, that we term the long-stop condition, where we used the same position, velocity, force and mean activation costs as in the stop model, but applied for reaching movements of 28 cm. The concept of the long-stop model is to compare the actual hit behaviour, executed through a different controller, with "cheating" behaviour where the same stop movement is performed to an imaginary target, located beyond the actual target, resulting in non-zero velocity at the actual target, and thus appearing as a hit movement. For all three conditions we then induced virtual target perturbations by shifting a target laterally by 2 cm at every time step from movement onset to movement end. With these simulations we obtained one continuous feedback response profile per condition, showing a dependency of feedback response intensity on time-to-target (Fig 3B and 3C). This feedback response profile is characteristic of the particular movement control policy associated with the movement goal, as it is maintained even if the kinematics of the movement change (Figure 8 in [26]).

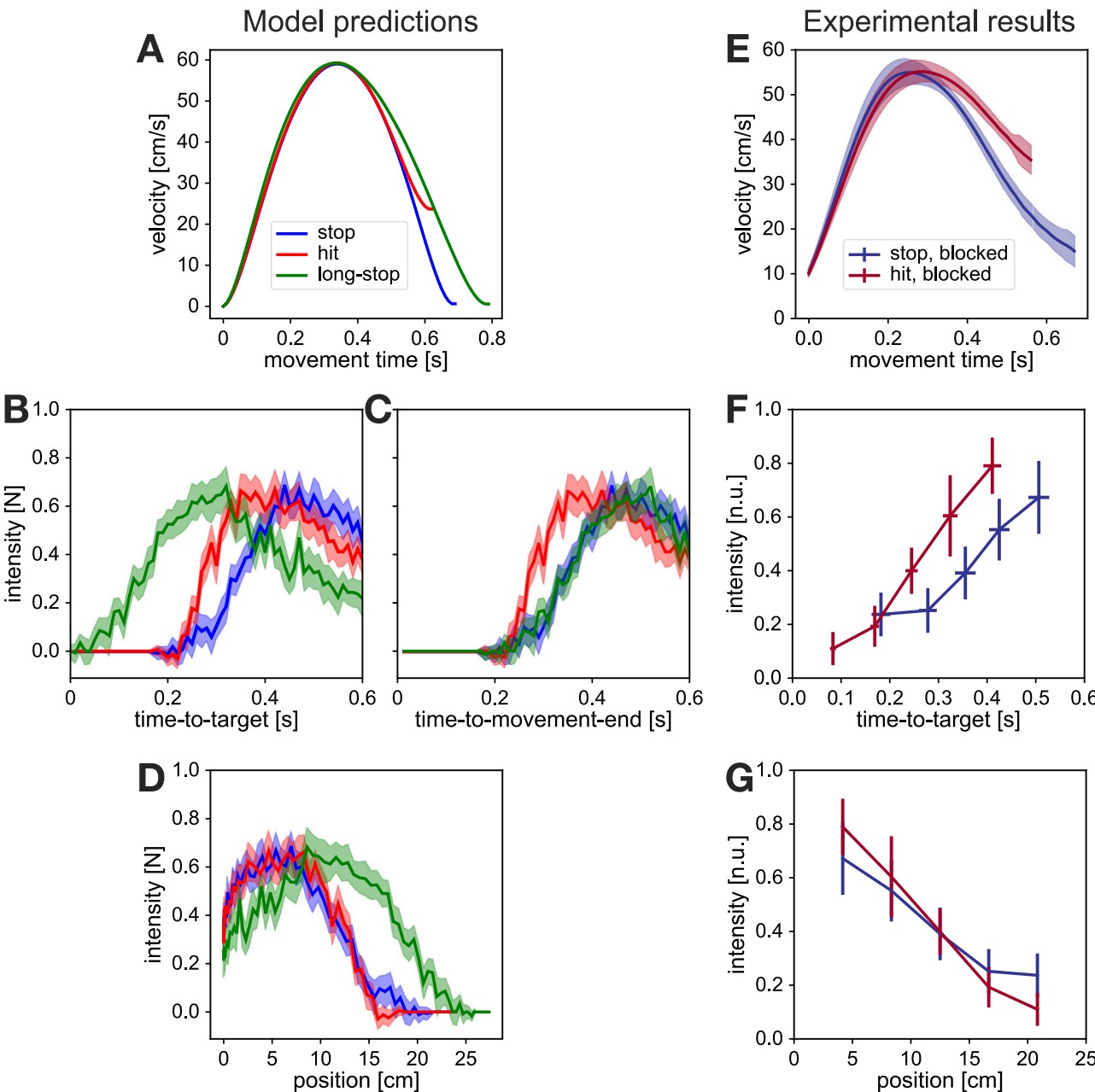

**Fig 3. Comparison of OFC model predictions and experimental results. A**. Simulated kinematics of stop, hit and long-stop conditions. Stop and hit conditions produce matching kinematics that only deviate shortly before movement end. The long-stop condition is a control simulation, that matched the kinematics of the hit condition for the duration of the hit movement, but was achieved with the same stop controller. **B**. Simulated feedback intensities as a function of time-to-target and **C**. time-to-movement-end. Simulations predict a faster increase of response intensities for hit condition than for stop condition. As the long-stop condition is simulated via a longer (28 cm) movement, the time-to-target represents a time until the simulated movement crosses a point of 25 cm distance (the target distance). For hit and stop conditions, time-to-target and time-to-movement-end are identical. When expressed against time-to-movement-end, long-stop produces matching responses to the stop condition, as the feedback controller used for these movements is identical. With respect to the time-to-target, long-stop responses are time-shifted from the stop responses. **D**. Simulated feedback intensities as a function of the position. Stop and hit simulations with these particular kinematics produce matching feedback intensity profiles when expressed against position, even if the feedback controllers are different. In contrast, the long-stop simulation with a feedback controller matching that of the stop condition still produces different intensity profile against position. Shaded areas in simulated traces represent 95% confidence intervals for simulated results. **E**. Velocity profiles of participants in blocked stop and blocked hit conditions. The profiles match the task requirements. **F**. Normalised feedback intensity profiles of participants in blocked stop and blocked hit conditions, expressed against time-to-target. Participants produce stronger responses at matching time-to-target in the hit condition, consistent with simulation results for hit and stop. **G**. Normalised feedback intensity profiles of participants in blocked stop and blocked hit conditions, expressed against position. Participants produce matching responses within hit and stop conditions, supporting model simulations for stop and hit conditions, and not stop and long-stop. Error bars in experimental results represent 95% confidence intervals.

Even with similar simulated kinematics, that deviate from each other only in the last portion of the movement, the OFC model predicts striking differences in the control policies for stop and hit conditions (Fig 3B and 3C, blue and red traces) or hit and long-stop conditions (Fig 3B and 3C, red and green traces) when expressed against time-to-target. On the other hand, when expressed against position, even different controllers (hit vs stop) show no differences in feedback intensities, while identical control (stop vs long-stop) exhibit clear differences (Fig 3D). Among other things, these results point out limitations of position as a dependent variable in determining the changes of control policies, and provide yet additional support for time-to-target.

Our models make a few predictions for the behaviour of human participants. First, it challenges the classic assumption that visuomotor feedback response profiles are always bell shaped, if probed at evenly spaced locations or movement times. Instead, we propose that the bell-shaped feedback response profiles are consequential to the specific kinematic values imposed by the experiments, and other, for example monotonically decreasing intensity profiles, are also possible with faster movements (Fig 3D). Second, our simulations also make predictions on relative differences between the feedback intensity profiles in stop and hit conditions. Particularly, we expect the hit condition to produce stronger responses than the stop condition for short times-to-target, with this relationship inverting for long times-to-target if the two types of movements require different feedback controllers (Fig 3B and 3C). Note, that while in previous studies it is typical to compare such response profiles in terms of perturbation onset location, here no difference between hit and stop is predicted in position-dependent profiles (Fig 3D).

## Human control policies match model predictions in hit and stop conditions

In order to compare the behaviour of our participants to the model predictions, we first analysed our results from the blocked schedule of the experiment. Here every participant has completed a block of 416 trials of hit condition and another block of 416 trials of stop condition, with the order counterbalanced across all participants. Our experimental results qualitatively match the predictions of our model. First, participants successfully differentiated between the kinematics of the hit and the stop condition, with both types of movements resulting in matching early and peak velocity ($v_{peak,stop}$ = 58.9 cm/s, $v_{peak,hit}$ = 58.1 cm/s), but with differences towards the end of the movement such that the velocity at the target is higher for the hit condition (Fig 3E). Specifically, in the hit condition participants produced movements with average velocity at the target of 38.5 cm/s, while successfully stopping at the target in the stop condition. In addition, similar to the model simulations, movements in the hit condition were of slightly shorter duration (630 ms vs 700 ms).

Qualitatively, the experimental feedback responses also match the model predictions (Fig 3F and 3G). First, due to relatively fast reaching velocities in our experiment, as well as the lack of maintained perturbations, all perturbations were induced at short times-to-target (under 550 ms). For comparison, in our previous study [26] perturbations were induced at times-to-target that ranged between 300 ms and 1000 ms, with peak feedback intensities recorded for perturbations with time-to-target at 400 ms. Second, both our data and the model produce feedback intensities at short times-to-target that are higher for the hit condition than for the stop condition, even in movement segments where the kinematics are otherwise similar. In addition, we also observe no learning effects within this regulation, as the relative behaviour across conditions is present in the first few blocks of the study, and remains throughout the entire experiment (S1 Text). Importantly, we do not fit the model to match the data, but

instead use it to qualitatively describe the relative regulation of stop and hit conditions. As such, matching features between the intensity profiles of the model (Fig 3B–3D) and the data (Fig 3F and 3G) imply that similar computational mechanisms may be in action. Finally, our results also indicate that participants utilise different feedback controllers for the hit and stop conditions, as the experimental results for the blocked hit condition match the model simulations of the hit, rather than the long-stop condition.

## Human participants utilise contextual switching of feedback controllers

In the previous sections we established the differences between the baseline control policies of hit and stop conditions. Here, we test how these policies change when the exposure to these conditions is no longer blocked. For example, it is natural in our daily activities to continuously switch between tasks, rather than doing a single task for many repetitions before switching to a new task. However, the question remains, how switching between different tasks affects the underlying feedback control policies. To test this, in the second half of the experiment we presented our participants with the same two types of movements (hit and stop), but now with the conditions randomly mixed across trials, instead of being presented in two separate blocks. As such, we could test for one of two possible outcomes:

1. Control policies for stop and hit movements in the mixed schedule match respectively the control policies in the stop and hit movements in the blocked schedule (Fig 1D). Such an outcome would indicate that participants are able to easily switch between different control policies (at least within consecutive trials).

2. Control policies for stop and hit movements in the mixed schedule do not match with the respective baseline policies, indicating interference when switching among multiple conditions (Fig 1C).

While both outcomes have previously been discussed from the sensorimotor adaptation perspective, to our knowledge they have not yet been demonstrated for feedback control.

Our participants successfully produced the movements required in the experiment (Fig 4A). Particularly, we observed clear distinctions in the terminal velocity between the hit and stop conditions, independent of the experimental schedule (blocked or mixed). A two-way repeated-measures ANOVA showed a significant main effect on condition (hit or stop, $F_{1,13}$ = 544.2, $p \ll 0.001$), but no significant main effect on experiment schedule (blocked or mixed, $F_{1,13}$ = 0.710, $p$ = 0.42) or schedule/condition interactions ($F_{1,13}$ = 0.681, $p$ = 0.42). In addition, a complementary Bayesian repeated-measures ANOVA analysis showed similar results, with a very strong effect [53] of condition (hit or stop, $BF_{incl}$ = $1.6 \times 10^{25}$), and with a tendency towards no effect of schedule (blocked or mixed, $BF_{incl}$ = 0.379), or condition/schedule interaction ($BF_{incl}$ = 0.409). A similar analysis for peak velocities showed a significant main effect of condition (hit or stop, $F_{1,13}$ = 5.94, $p$ = 0.03; although $BF_{incl}$ = 1.12 indicates not enough evidence to either reject or accept the null hypothesis) and condition/schedule interaction ($F_{1,13}$ = 19.3, $p \ll 0.001$; $BF_{incl}$ = 32.6), but not on schedule (blocked or mixed, $F_{1,13}$ = 1.52, $p$ = 0.24; $BF_{incl}$ = 0.56 shows a weak tendency towards accepting null hypothesis). The Holm-Bonferroni corrected post-hoc analysis for the interaction term revealed that participants produced slightly faster movements in the mixed-hit condition, with the peak velocities matching otherwise.

We examined the evolution of the experimental visuomotor responses as a function of perturbation onset position or onset time across the four different conditions (Fig 4C and 4D). When expressed against either position or time, the visuomotor intensity profiles do not show the classical bell-shaped profile where strongest responses occur in the middle of the

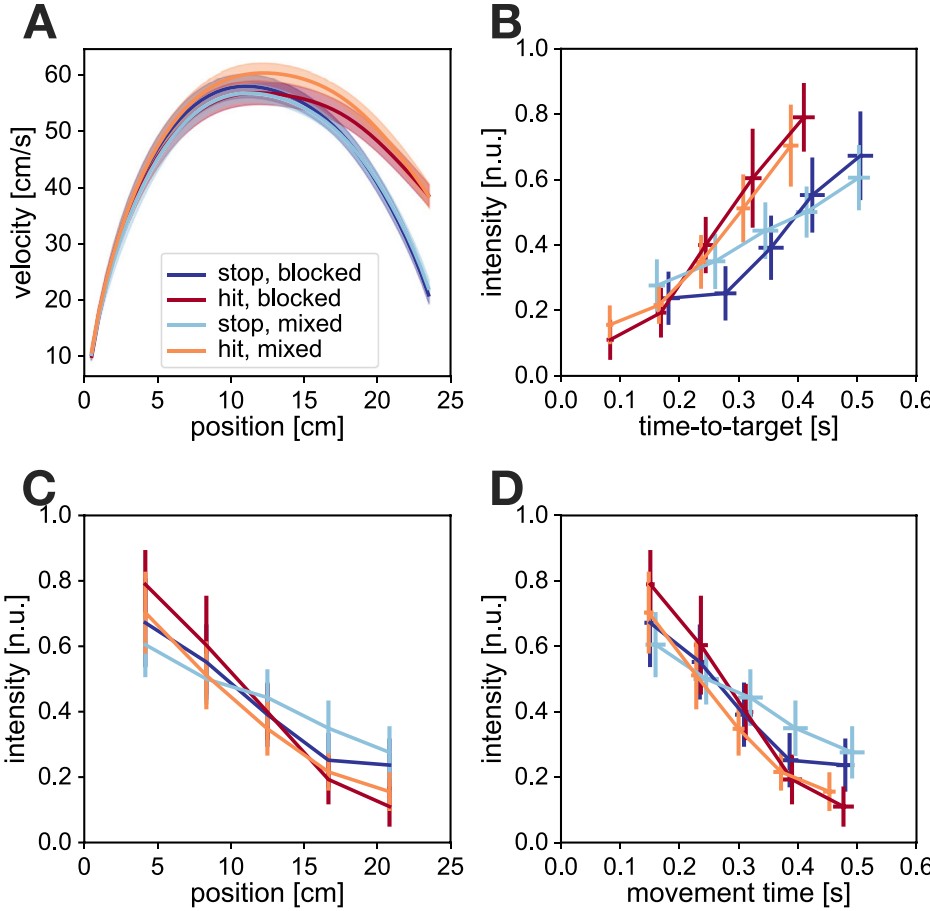

**Fig 4. Experimental results of stop and hit conditions in both blocked and mixed schedules. A**. Velocity profiles against position. Both stop conditions and both hit conditions produce respectively similar velocity profiles, showing that participants successfully performed the task in the mixed schedule. **B**. Normalised feedback response intensities represented as a function of time-to-target. Hit and stop movements in the mixed schedule demonstrate differences when expressed against time-to-target, that match the differences between hit and stop conditions in the blocked schedule. This supports the hypothesis of contextual controller switching between multiple task-specific controllers. **C**. Normalised feedback intensities in all four conditions show no differences when expressed against position or **D**. movement time at perturbation onset, as predicted by the OFC simulations. This questions the appropriateness of position or movement time as the reference frames in which to compare multiple feedback controllers. Error bars and shaded areas indicate 95% confidence intervals of the mean.

movement and are reduced towards the beginning and end. Instead, our participants produced the strongest responses for the earliest perturbations, induced at 1/6 of the total forward movement, with further responses decaying in intensity as perturbations occurred closer to the target. Moreover, we observed no significant differences in visuomotor responses across the different conditions and schedules. Three-way repeated-measures ANOVA with condition (stop or hit), schedule (blocked or mixed) and perturbation location (5 levels) as main factors showed no effect of condition ($F_{1,13} = 0.486$, $p = 0.50$; $BF_{incl} = 0.238$ shows substantial evidence towards no effect), schedule ($F_{1,13} = 0.096$, $p = 0.76$; $BF_{incl} = 0.142$ shows substantial evidence towards no effect) or condition/schedule interaction ($F_{1,13} = 0.657$, $p = 0.43$; $BF_{incl} = 0.305$ shows substantial evidence towards no effect). While we observed a significant main effect of the perturbation location ($F_{2.9,37.7} = 61.2$, $p \ll 0.001$ after Greenhouse-Geisser sphericity correction; $BF_{incl} = 9.3 \times 10^{36}$), such an effect was expected due to the temporal evolution of feedback responses. In addition, we observed a significant interaction between perturbation onset

location and the condition ($F_{2.1,27.0} = 6.26$, $p = 0.005$ after the sphericity correction; $BF_{incl} = 6.86$), however a Holm-Bonferroni corrected post-hoc analysis on the interaction term did not indicate any meaningful interaction effects, with none of the significant interactions appearing at the same perturbation onset location. Finally, the remaining interactions of schedule/perturbation ($F_{2.6,33.9} = 2.67$, $p = 0.07$ after Greenhouse-Geisser sphericity correction; $BF_{incl} = 0.289$) and condition/schedule/perturbation ($F_{2.8,36.8} = 0.233$, $p = 0.86$ after Greenhouse-Geisser sphericity correction; $BF_{incl} = 0.075$) showed no significant effects. Thus, as a whole our analysis indicates that the feedback controllers could not be differentiated when expressed as a function of the position within the movement.

When expressed against time-to-target, the visuomotor feedback responses show decreasing feedback intensities with decreasing time-to-target, with responses virtually vanishing when the time-to-target approaches zero (Fig 4B). This behaviour is consistent with our previous models describing the time-gain relationship [26]. In addition, we observe stronger increases in visuomotor feedback intensity with increasing time-to-target for the hit condition compared to the stop condition, in both blocked and mixed schedules. Such regulation was previously predicted by our time-to-target OFC model (see Figure 9C in [26]) for short times-to-target. Finally, we also observe a qualitative match between the two stop conditions (mixed and blocked) as well as between the two hit conditions (mixed and blocked), suggesting first evidence of rapid feedback controller switching in the mixed schedule. This finding holds equally well in trials immediately after a condition switch, as well as after the trials of the same movement condition (S2 Text). Qualitatively the increase of visuomotor response intensities with time-to-target for our specific results could be well described by a line function for each of the four combinations of condition and schedule. In order to get a quantitative estimate of the differences between the conditions we performed a Two-way ANCOVA analysis of visuomotor response intensity, with schedule and condition as the two factors, and time-to-target as the covariate. The results showed a significant main effect of condition (hit or stop, $F_{1,275} = 24.8$, $p \ll 0.001$; $BF_{incl} = 9.46 \times 10^3$), and time-to-target ($F_{1,275} = 222.8$, $p \ll 0.001$; $BF_{incl} = 1.04 \times 10^{33}$), but no effect of the experimental schedule (blocked or mixed, $F_{1,275} = 0.098$, $p = 0.75$; $BF_{incl} = 0.138$) or of schedule/condition interaction ($F_{1,275} = 1.06$, $p = 0.30$; $BF_{incl} = 0.304$ shows tendency towards no effect). Such results indicate that we can successfully separate the two different controllers when expressing their feedback response intensities (or their gains) against the time-to-target. Furthermore, we also show that such differences are only present when comparing the controllers for different tasks, and are not dependent on the presentation schedule of these tasks. Thus, we demonstrate that our participants successfully selected an appropriate controller for a hit or a stop task, even in a schedule where the task could change on consecutive trials.

## Discussion

In this study we have demonstrated that humans are capable of rapid switching between appropriate feedback controllers in the presence of different contextual cues. Specifically, our participants show systematic differences in feedback responses when performing hitting movements, compared to reach-and-stop movements. Moreover, the same systematic differences are present, both when the two tasks are performed in isolation (blocked schedule), or when rapidly switching from one task to the other (mixed schedule), showing that these differences are evoked within a single trial, and not gradually adapted. Finally, these feedback responses are also well matched with the optimal feedback control predictions for these responses in hit and stop tasks, further reinforcing accumulating evidence of optimality principals in the feedback control of human movements.

In order to gain insight into computational mechanisms that are employed when humans switch between hit and stop conditions, in this study we formulate our hypothesis through normative modelling [37, 54–58]. Such an approach compares the behavioural experimental data to the results simulated computationally through a known bottom-up design. In turn, any mismatch between the data and the model rules out the mechanism, while matching behaviour provides support for the likelihood of such a mechanism. Specifically, here we simulate three different types of control movements: stop movement, where a point mass is stopped at a target 25 cm away from the start position; hit movement, where the point mass is instead brought to the same target with nonzero terminal velocity; and a long-stop movement, with similar kinematics to the hit movement within the 25 cm segment, generated by a stop movement to a secondary virtual target at 28 cm distance. The hit and stop simulations differed in the implementation of the feedback controller, with the state dependent costs for the terminal velocity and terminal acceleration reduced by a factor of 50 for the hit condition. As a result, the two models inherently simulate the behaviour that is achieved via different controllers. On the other hand, the long-stop condition was simulated by using the same controller as the stop condition, but to a target at 28 cm instead of 25 cm. Consequently, such a movement still maintained a non-zero velocity at 25 cm, virtually simulating a hit-like movement. Notably, in order to better match the kinematics of a long-stop movement to the kinematics of the hit and stop movements, we temporally modulated the activation cost $R$ of the long-stop controller, which we have previously shown does not affect the overall feedback responses in terms of time-to-target [26]. In general, while kinematics of hit and long-stop models matched well, the two simulations predicted very different feedback response profiles when expressed both against time-to-target and against position. Finally, the responses of our participants in the hit condition matched better with the model simulation of the hit condition, rather than the simulation of the long-stop, providing evidence that humans use different feedback controllers for different tasks.

Principles of contextual switching have been extensively studied in the context of feedforward adaptation [30, 31, 34, 59–62]. While these cues vary in effectiveness [30, 59] and are typically considered as relative weightings of multiple feedforward models [63], strong dynamic cues such as differences in follow-through [35, 36], lead-in [33, 61], or visual workspace [31, 32] can effectively separate the feedforward models. As multiple recent papers have demonstrated that voluntary (feedforward) and feedback control likely share neural circuits [24, 39, 64–67], it is reasonable to believe that similar contextual regulation would also be present in feedback control. However, studies that have shown this parallel changes in the feedback responses to the learning of the feedforward dynamics, either examined before and after adaptation to novel dynamics [24, 64, 68, 69], or during the process of adaptation [19, 70–72], meaning that the they could not distinguish between the slow adaptation of the feedback controller to each condition or the rapid switching between two controllers. Moreover, other studies have suggested that feedforward and feedback controllers are learned separately [73, 74] and may even compete with one another [75], suggesting that these share different neural circuits and may have different properties. In this study we showed that in the mixed schedule, where the task goal unpredictably switched between hit or stop tasks on consecutive trials, participants evoked different control policies for each task. Furthermore, these policies, evoked within mixed schedule, well matched with the respective policies in the blocked schedule, suggesting that they were not only different from one another, but also appropriate for each task, showing the strong separation of the two contexts. While this is not unexpected, as the two hit and stop tasks are significantly different in their dynamics and thus should act as a strong contextual cue, one important result is that we demonstrated that the context regulates the feedback, and not only feedforward control. Finally, our results are also consistent with the

accumulating evidence of the shared relationship between feedforward and feedback control in human reaching.

One reason why contextual effects on feedback control have not been broadly studied, is that it is difficult to quantify what really constitutes a change in feedback control policy. For example, we can trigger responses of different magnitudes by changing the size of the perturbation [41, 42], inducing perturbations at different positions [26, 40] or at different times [20, 37, 76]. However, computationally such differences in response intensity can be achieved within the same optimal feedback controller without ever changing control parameters [38]. On the other hand, experimental tasks, presented in some of these studies, e.g. reaching towards narrow, wide or long targets, inherently require different feedback controllers. Specifically, assuming a similar controller to the one we present in this work, a wide target implies a reduced $\omega_{p,t}$ weight in x-axis compared to the narrow target, thus leading to a different optimum of control matrix $L$ (Eqs 2 and 3). Indeed, human responses in tasks where the target structure changed (either by shape or by the presence of obstacles [42, 44, 45]) were consistent with the OFC predictions of two independent controllers [42]. In this article we present two tasks that also require different feedback controllers, but achieve that while maintaining the target shape. Instead, we invoke different controllers by modulating the task requirement of either stopping at the target, or hitting through it. In addition, by combining an OFC model predictions with our previous work, showing that the time-to-target is a strong predictor of the feedback intensity in optimal control tasks [26], we not only show that the human behaviour is consistent with two independent controllers, but also that it cannot be explained by one controller. Specifically, we simulate the behaviour either by recomputing the controller $L$ (Fig 3B–3D, hit and stop), or by updating the state estimate $\hat{x}$ and using the same controller $L$ (Fig 3B–3D, stop and long-stop), to compare with the experimental results (Fig 3F and 3G). These results show that humans indeed change their control policies when the task goal (e.g. hit or stop) changes. Thus, by combining behavioural results with normative control models we can clearly identify that it is specifically the change in control, and not other mechanisms, that is responsible for the regulation observed in the experimental data.

Previous studies have demonstrated that visuomotor feedback intensity profiles are roughly bell-shaped along the movement—low at the beginning and the end, and peaking in the middle [26, 40]—leading to assumptions that these gains might parallel the velocity [39, 67]. Our simulations and experimental results (Fig 3D and 3G) demonstrate that this bell-shape profile is not fixed, and that other profiles are possible. In our previous work, we established a robust relationship between the visuomotor feedback intensities and time-to-target, demonstrating that time-to-target is the fundamental variable that modulates the responses, given that the task goal (and thus the feedback controller) remains the same [26]. This means that the bell-shaped profile is simply a by-product of a specific timing of perturbations, and is not regulated by their onset location. As a consequence, the shape of these feedback intensity profiles can be modulated away from the bell-shaped profile by changing movement speed, target distance or acceleration profile. Such results illustrate possible caveats in the experimental paradigms of motor control: historically, some of the task requirements have been largely consistent, particularly in terms of reaching distance, reaching speed or duration. This may result in some measured behavioural outcomes being specific to these kinematics or conditions rather than representing the general features of the motor control system. Thus, while we do not advocate for routinely altering the standard experimental and analytical methods, it is worth considering the specific biases that such methods may contribute to a given study.

One popular way of looking at the visuomotor responses in humans is how they vary with position in a movement. Indeed, numerous studies either analyse the evolution of responses against position [26, 39, 40], or induce perturbations based on a fixed position [7, 9, 25, 41,

49–51], with the expectation that these perturbations induce similar responses unless the control changes. For example, [9] demonstrated different feedback responses, induced at a matched position in movements towards different targets. While we believe that these different target properties indeed suggest different feedback controllers, such a distinction cannot be reliably tested with only one perturbation, matched by position. Our results clearly demonstrate the limitations of position as the main variable to probe such control. On one hand, even with similar kinematics for the majority of the movement, simulations of stop and long-stop movements predict radically different responses at matching positions (Fig 3D), despite the fact that these are generated with identical controllers. On the other hand, different controllers for hit and stop conditions still produced roughly matching feedback responses at the same position, consistent with the experimental data (Fig 3D and 3G). In contrast to position as the main variable, OFC simulations in both this study and our previous work [26] show that the same controller, when expressed against time-to-target, produces matching response profiles, independent of other kinematic factors such as movement velocity or position of the perturbation onset (Fig 3B and 3C). Furthermore, different controllers, such as hit and stop, produce feedback responses with systematic differences when expressed against time-to-target, exactly as demonstrated by our participants. Thus, we propose that time-to-target is the better reference frame for comparing feedback responses.

In this study we have raised two alternative hypotheses about the regulation of feedback controllers within the mixed schedule. The first possibility is that the feedback control gradually adapts to a given task over a few consecutive trials, similar to the feedforward control during learning of a force field or visuomotor rotation. If such adaptation was true, we expect different feedback intensities between the hit and stop conditions in the blocked schedule as the controller has enough trials to reach steady-state behaviour. However, in the mixed schedule the controller would drift between the equilibrium of hit and stop conditions, producing similar responses for mixed hit and mixed stop conditions. Note that even in such a case where only a single controller is performing both hit and stop trials, we would not necessarily expect any effects on the kinematics or the participant's ability to complete the task. Instead, due to the feedback nature of the control, a sub-optimal controller would still complete the movement, but produce sub-optimal (e.g. more costly) responses in the presence of external disturbances. The second possibility is that an appropriate controller is selected before each movement based on the provided context, allowing immediate switching between tasks. In this case, the feedback intensity profiles would match for the same task, regardless of the schedule of their presentation. That is, we expect to see similarities between both hit conditions, as well as between both stop conditions, but differences between any two hit and stop conditions. Our experimental results strongly support the latter option, as we not only observe differences between mixed hit and mixed stop conditions, but also observe their respective match with the blocked conditions. While our results do not rule out the adaptation of feedback controllers in general, we do demonstrate that different optimal controllers can be rapidly selected and switched between for familiar tasks.

One important aspect of the relationship between feedforward and feedback control is that modulating one of them should affect the behaviour of the other. Indeed, previous work has demonstrated that human participants changed their feedback gains after adapting their feedforward models to novel dynamics [11, 19, 24, 64, 68–72]. However, an adapted movement in the force field typically produces kinematics that are similar to those in baseline movements, suggesting that such change of gains is achieved at matching times-to-target, and with the same task goal. Thus, our proposed framework that the relation between feedback intensities and time-to-target is unique for a unique controller would predict that the feedback gains would remain unchanged. As a result, we can not directly explain this change of control gains,

unless the feedback controller somehow changes during adaptation. One factor driving such a change is that adapted movements in the force field are more effortful than baseline movements, due to additional muscle activity required to compensate for the force. An increased effort in the context of OFC simulations would thus increase the model activation cost $R$, resulting in a change of optimal feedback gains and intensities at matching times-to-target. In addition, the presence of a force field likely influences the biomechanics of the movement (particularly the muscle viscosity $b$), changes the state transition due to the external dynamics (via state transition matrix $A$), and updates the state uncertainty [77], resulting in the same controller being applied to a different control plant, and thus producing different responses. Moreover, if the controller is optimised to to this new control plant, adaptation will inevitably require a new feedback controller. Therefore, such changes in feedback control are expected, even though conventionally it appears that the task goal remains the same after adaptation to the novel dynamics.

Even though many recent studies use force channel trials [52] to accurately measure the visuomotor feedback responses [7, 11], often these brief perturbation trials are complemented with maintained perturbation trials [19, 26, 41, 46–48, 50, 51]. This is because brief perturbations within a channel trial are task-irrelevant, and can be ignored without compromising the task, whereas maintained perturbations strengthen these responses as they require an active correction for the participant to reach the target. However, we have recently shown that these maintained perturbations also force a non-trivial extension of the movement duration compared to the non-perturbed movement, and thus complicate the relationship between the perturbation onset location and the time-to-target. Hence, in order to consistently evaluate the control behaviour and its relation to the time-to-target, here we deliberately chose to only induce perturbations within the force channels and not to include the maintained perturbations. Although this generally decreases overall feedback intensities, our participants produced clear responses that exhibited the temporal evolution as predicted by the OFC model simulations.

Another possible limitation of using channel trials to probe the feedback control is the potential interference of the stretch reflex. Specifically, small forces produced on the hand by the channel could set on the feedback corrections [78] that superimposed onto the measured visuomotor responses. However, as the onset of the force channel occurs prior to any movement of the participant, long before the time of the visual perturbation, the channel onset will not produce a stretch reflex response timed to the visual stimuli. More importantly, such effects, as well as any other corrections to the channel onset would be present in all channel trials (including zero-perturbation trials), and thus would cancel-out in the net feedback responses, as they would not depend on the direction of the visual perturbation. As a result, it is unlikely that the force channels introduced systematic effects into our recorded visuomotor feedback signal. Similarly, we also saw no behaviour differences between simulations of free movements, that we presented in our results, and analogous simulations of matching duration movements in channel trials (S1 Fig).

Most studies of motor learning study contextual switching in conjunction with dual adaptation by introducing participants to novel force fields or visuomotor rotations, for which they do not have pre-existing feedforward controllers. In turn, we typically see slow, simultaneous adaptation to applied perturbations, as well as context-induced switching of the memories after the transient learning phase is over. Importantly, for familiar tasks this switching is evoked immediately, without the need to re-learn the dynamics again on re-exposure. This is clearly seen on the second or later days after adapting to dual force fields [35, 79]. In this study, our main goal was to demonstrate that such contextual switching is also possible for feedback controllers, rather than to demonstrate gradual adaptation. Therefore, in our

experimental design we consciously selected two tasks (stop or hit) that were not novel to our participants. While there remains a possibility that due to the dynamics of the vBot environment both tasks were different to stop or hit movements outside of the lab, and thus novel to participants, we always started our studies with the blocked schedule, and only then followed with the mixed schedule to make sure that the different baseline controllers are already available to our participants. An interesting control would be to first test the participants in the mixed schedule, followed by the blocked schedule. However, we believe such control would mainly test whether the two task choices were novel to participants or not, which is not the focal point of our study.

In summary, here we again demonstrate that time-to-target (which could be considered as one form of urgency) [20, 26, 39, 76, 80], and not position or velocity, act as a primary predictor for the feedback response intensity when the task goal is fixed. Moreover, when comparing multiple tasks, the time-to-target reference frame consistently separates the feedback control policies for these tasks—an outcome that fails when comparing two different controller gains within the position reference frame. While position within the movement, and velocity at the time of a perturbation, definitely influence the controller responses, our results clearly demonstrate that the effect of these variables on overall control may be somewhat exaggerated in the previous literature. For example, our participants produced temporal evolution of the responses to visual perturbations that neither paralleled the velocity, nor showed the typical variation with position (with peak responses achieved mid-movement), but could be explained by the time-to-target dependency that was predicted by OFC. In addition, participants were able to switch their feedback controller from one trial to another, demonstrating the principle of contextual switching for feedback control. Such switching, well known in feedforward control, further reinforces accumulating evidence of the shared connections between feedforward and feedback control. Most importantly, our results demonstrate that the visuomotor feedback control in humans not only follows the principles of optimal control for a singular task, but also selects an appropriate controller for that task upon presenting the relevant context.

## Methods

### Ethics statement

The study was approved by the Ethics Committee of the Medical Faculty of the Technical University of Munich. All participants have provided a written informed consent before participating in the study.

### Experimental setup

Fourteen right-handed [81] human participants (age 21–29 years, 5 females) with no known neurological diseases and naïve to the purpose of the study took part in the experiment. Participants performed forward reaching movements either to a target (stop condition) or through the target (hit condition) while grasping the handle of a robotic manipulandum (vBOT, [82]) with their right hand, with their right arm supported on an air sled. Participants were seated in an adjustable chair and restrained using a four-point harness in order to limit the movement of the shoulder. A six-axis force transducer (ATI Nano 25; ATI Industrial Automation) measured the end-point forces applied by the participant on the handle. Position and force data were sampled at 1 kHz, while velocity information was obtained by differentiating the position over time. Visual feedback was provided via a computer monitor and mirror system, such that this system prevented direct vision of the hand and arm, and the virtual workspace appeared in the horizontal plane of the hand (Fig 5A). The exact timing of any visual stimulus presented to the participant was determined from the graphics card refresh signal.

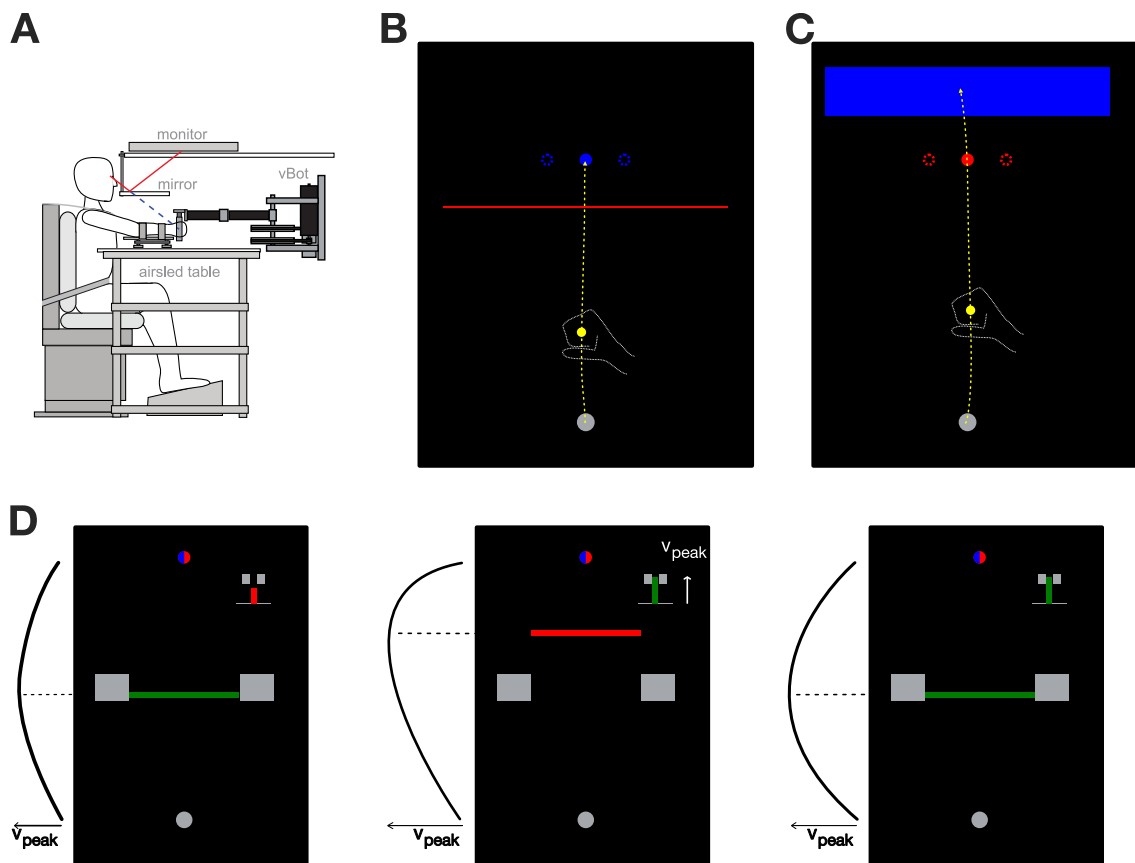

**Fig 5. Experimental setup. A**. Participants controlled a yellow cursor by moving a robotic handle. The cursor was projected via a screen-mirror system directly into the plane of the participant's hand. Figure copyright 2008 Society for Neuroscience. **B**. Stop condition. Participants were instructed to reach with the cursor through a red line and stop within the blue target. Target perturbations were occasionally induced via target jumps of 2 cm laterally. **C**. Hit condition. Participants were instructed to reach through the red target and stop within the blue area. Target perturbations (2 cm laterally) were again induced on random trials. **D**. Visual feedback was presented after each trial. Participants were shown the workspace with the start position and the target still present. In addition, two indicators were displayed. A bar chart at the top-right part of the workspace scaled proportionally with the absolute peak velocity, and was green if the velocity was within the required range as indicated by two grey brackets. A horizontal bar indicating the actual forward location where this peak velocity was achieved was displayed between the start and target positions. This bar was green if the peak location matched the experimental requirements, indicated by two large rectangular blocks. If both location and peak amplitude criteria were successfully fulfilled, participants were rewarded with one point. If at least one of the two criteria was not fulfilled, the respective indicator turned red instead of green, and no point was provided. In both hit and stop experiments participants were instructed to move through the red workspace element and stop at the blue, and were rewarded with one point if they both intercepted the target and fulfilled both velocity requirements.

Participants controlled a yellow cursor (circle of 1.0 cm diameter) by moving the robotic handle. The centre position of this cursor in the virtual workspace always matched the physical position of the handle. Every experimental trial was initiated when the cursor was brought into the start position (grey circle of 1.6 cm diameter), which was located 20 cm in front of participants' chest and centred with the body. When the cursor was within this start position, the circle changed from grey to white and the type of experimental trial was indicated by the presentation of a target. After a random delay, sampled from an exponential distribution with λ = 0.7 and truncated outside 1.0 s–2.0 s interval, a tone was played to indicate the start of the movement. If participants failed to leave the start position within 1000 ms after this tone, the procedure of the current trial was aborted and restarted.

Over the course of the experiment the participants were tasked to complete two types of movements: stop movements, where they were required to stop within the target (a circle of 1.2 cm diameter, located 25.0 cm in front of the start position) (Fig 5B), or hit movements, where they had to intercept the target without stopping, and instead stop in a designated stopping area (a blue rectangle, [width, height] = [15 cm, 4 cm], centred 5 cm beyond the target) (Fig 5C). The reaching movement was considered complete once the centre of the cursor was maintained for 600 ms either within the area of the target in stop trials, or within the stopping area in the hit trials. In addition, if the movement duration was longer than 4.0 s, the trial was timed-out and had to be repeated. After each trial, the participant's hand was passively moved back to the start position by the vBOT, while the feedback of the previous trial was provided on screen (Fig 5D). All movements were self-paced, with short breaks provided every 208 trials, and a longer break (5–10 minutes) provided at the half-way point of the experiment.

## Experimental paradigm

Participants performed reaching movements in four conditions—blocked stop, blocked hit, mixed stop and mixed hit—that were part of a single experiment. Across these conditions, participants were required to either reach to the target and stop there (the stop conditions), or to reach through the target and stop in the designated stopping area (hit conditions). In order to easily cue the distinction between the hit and stop conditions, the two types of trials had small visual differences. For the hit condition participants were presented with a red target (a red circle of 1.2 cm diameter) and a rectangular blue stopping area of dimensions 15 cm by 4 cm, centred 5 cm beyond the target (Fig 5C). For the stop condition participants were presented with a target that was otherwise identical to the target in hit condition, but was blue in colour, and with a horizontal, 15 cm wide red line, that was placed 3 cm before the target (Fig 5B). While this line had no functional interaction with the experiment, it allowed us to consistently instruct the participants to always perform reaching movements so that they intercept the red element in the workspace, and stop within the blue element.

In order to probe the visuomotor feedback responses of participants, during some reaching movements we briefly perturbed the target by shifting it 2.0 cm laterally for 250 ms before returning back to the original position (Fig 2A). These perturbed trials were always performed within the virtual mechanical channel, where participants were free to move along the line between the start position and the target, but were laterally constrained by a virtual viscoelastic wall with stiffness of 2 N/m and damping 4000 Ns/m [7, 40, 52]. As the perturbations were always task-irrelevant, this channel therefore did not obstruct participants to complete the trial. However, as participants still produced involuntary feedback responses due to the target shift, the virtual channel allowed us to record the forces that participants produced due to the perturbations and measure the intensities of the visuomotor feedback responses.

For each type of movement (i.e. hit or stop) there was a total of 11 different perturbations. Ten of these perturbations were cued during the reaching movement as participants crossed one of the five perturbation onset locations, equally spaced between the start position and the target position (4.2, 8.3, 12.5, 16.7, and 20.8 cm from the centre of the start position). At all of these five locations the target could either shift to the left or to the right. In addition, one zero-magnitude perturbation was also included, where the movement was simply performed within the channel without any target shift in order to probe the force profile of the natural movement. Finally, in addition to the perturbation trials we also included non-perturbed trials where participants simply reached towards the target without any target perturbation and without the virtual channel constraining the hand.

In order to present the different perturbations in a balanced manner, we combined different types of trials in blocks of 16 trials. One block of 16 trials contained 11 perturbed trials (5 perturbation onset locations x 2 directions, plus one neutral movement in the force channel), and 5 non-perturbed movements without the force channel. Each of the four experimental conditions consisted of 26 such blocks, with the order of trials fully randomised within each block, resulting in 416 trials per condition and 1664 trials overall.

In the first half of the experiment, participants were always presented with the two blocked-design conditions (blocked hit and blocked stop), with the order of the conditions balanced across the population of participants. That is, each participant started with 416 trials of stop trials, followed by 416 hit trials or vice-versa. In the second half of the experiment, the two final conditions—mixed hit and mixed stop—were presented in a pseudo-random order within the same blocks. While individual trials within mixed conditions were identical to the individual trials within the respective blocked conditions, they were now presented in a pseudo-randomised order. Specifically, the remaining 832 trials were divided into 26 blocks of 32 trials, with each block consisting of 16 hit and 16 stop trials fully randomised within this block. Such randomisation resulted in a percentage split where 52% of trials were presented after a condition switch, 26% of trials were presented after exactly one trial of the same condition, 12%—after exactly two trials of the same condition, and larger clusters with diminishing frequency.

## Feedback regarding movement kinematics

In theory, the movements in hit condition could be interpreted as the movements, where the goal is to go through the via-point (the red target) and stop at the blue stopping area. As a result, such movements could simply be treated by participants as the stop movements with longer movement distance and a less restrictive target. Typically for such reaching movements, humans would produce a velocity profile that is bell-shaped, with peak velocity near the middle of the movement, and therefore further along the movement than in the stop condition. In order to avoid such differences and keep the velocity profiles comparable between the two conditions, we provided the task-relevant feedback on the velocity profiles, specifically the peak velocity and peak velocity location, to our participants (Fig 5D).

Independent of the experimental condition, participants were required to produce the movements with the peak velocity of 60 cm/s ± 8 cm/s, and the peak velocity location within 11.25 cm–13.75 cm movement distance (or 45%-55% of the distance between the start location and the target). The peak velocity was indicated as the small bar chart at the top-right of the screen, with the required velocity range indicated by two grey brackets. If the velocity target was matched, the bar chart turned green, otherwise it was red. Similarly, the peak velocity location was shown as a horizontal bar, centred around the movement distance where the peak velocity was reached. If this location was within the target range (also indicated by grey brackets), it was displayed as green, otherwise it was red. Participants were rewarded one point if both velocity requirements were successfully met, and the cursor intercepted the target during the movement.

## Data analysis and code availability

All data was pre-processed for the analysis in MATLAB 2017b: force and kinematic time series were low-pass filtered with a tenth-order zero-phase-lag Butterworth filter with 15 Hz cutoff and resampled at 1 kHz to account for an occasional missed sample during the signal recording. All subsequent analysis was performed in Python 3.9.4 and JASP v0.14.1 [83]. First, raw visuomotor feedback intensities were calculated from the force responses, recorded after the induction of a target perturbation. Specifically, for every perturbation trial we averaged the

lateral force response over a time window of 180 ms–230 ms after the onset of the perturbation, and subtracted a neutral force profile over the matching time window. This method and the particular time window has now been used in numerous studies to calculate the intensity of the early involuntary visuomotor feedback response [7, 25, 26, 40, 41, 50, 51]. As the direction of the response differed based on the perturbation direction, we reversed the direction of the intensities of responses to the leftward perturbations, so that positive intensities always indicate movements in correct direction, and grouped all intensities by the perturbation onset location. Second, we normalised mean feedback responses between 0 and 1 for each participant in order to avoid the group effect being biased towards participants with stronger responses. Finally, in our analysis the start of all movements was defined as the last time sample where the cursor is still within the area of the start circle, and the end of the movement was defined as the last time sample before the cursor enters the target circle. Time-to-target values were extracted from the data for every perturbation trial by subtracting the perturbation onset time from the movement end time.

In this article we provide two types of statistical analysis: the conventional frequentist statistics, as well as complementary Bayesian analysis that is presented as Bayesian factors [53], which instead of a simple hypothesis testing provides evidence for or against the null hypothesis. As a result, among other things, Bayesian analysis allows us to distinguish between accumulating evidence for the null hypothesis, and simply lacking evidence in either direction due to low power or small sample size.

All the Jupyter notebooks for the data analysis, pre-processed experimental data and statistical analysis conducted in this article are available at https://doi.org/10.6084/m9.figshare.17113904.v1.

## Computational modelling

In this study we formulated our initial hypothesis about the feedback control mechanisms in humans by first simulating the behaviour of the optimal feedback controller (OFC). Specifically, we used a finite-horizon linear-quadratic regulator framework—a relatively simple OFC that assumes perfect sensory input, as well as no control-dependent noise, while still being able to capture a significant part of the variance of human reaching movements [38, 84]. In order to model the feedback behaviour of our human participants, we first simulated virtual movements of a point mass with $m = 1$ kg, and an intrinsic muscle damping $b = 0.1$ Ns/m. This point mass was controlled in two dimensions by two orthogonal force actuators that simulated muscles, and regulated by a control signal $u_t$ via a first-order low-pass filter with a time constant $\tau = 0.06$ s. At time $t$ within the movement, such system could be described by the state transition model:

$$x_{t+1} = Ax_t + B(u_t + \xi_t), \tag{1}$$

where $A$ is a state transition matrix, $B$ is a control matrix, and $\xi_t$ is additive control noise. For one spacial dimension $A$ and $B$ are defined in discrete time as:

$$A = \begin{bmatrix} 1 & \delta t & 0 \\ 0 & 1 - b\delta t/m & \delta t/m \\ 0 & 0 & 1 - \delta t/\tau \end{bmatrix},$$

$$B = \begin{bmatrix} 0 \\ \delta t/\tau \\ 0 \end{bmatrix}$$

Finally, to simulate our model in discrete time we used the sampling rate $\delta t = 0.01$ s.

State $x_t$ exists in the Cartesian plane and consists of position $\vec{p}$, velocity $\vec{v}$ and force $\vec{f}$ (two dimensions each). The control signal $u_t$ is produced via the feedback control law:

$$u_t = -Lx_t \tag{2}$$

where $L$ is a matrix of optimal feedback control gains, obtained by optimising the performance index (also known as the cost function):

$$J = \sum_{t=0}^{N} x_t^T Q_t x_t + u_t^T R_t u_t = \sum_{t=0}^{N} \omega_{p,t}(\vec{p}_t - \vec{p^*})^2 + \omega_{v,t} \| \vec{v}_t \|^2 + \omega_{f,t} \| \vec{f}_t \|^2 + \omega_{r,t} \| u_t \|^2. \tag{3}$$

Here $x_t^T Q x_t$ and $u_t^T R u_t$ are two components of the total cost, known as state-cost and a control-cost respectively. In addition, $\omega_p$, $\omega_v$ and $\omega_f$ are position, velocity and force state cost parameters, $\vec{p^*}$ is a target position, $\omega_r$ is the activation cost parameter and $N$ is the duration of the movement, here required as a model input. Within the finite-horizon formulation, the cost parameters can be non-stationary and thus be different for every time-point. However, in our simulations we set $Q = 0$ for $t \neq N$, consistent with [37, 85].

In this study we simulate three different controllers that we call stop, hit and long-stop. While the stop and long-stop controllers are derived from the identical set of costs state-costs $Q$, they are used for slightly different movements (25 cm and 700 ms for stop, 28 cm and 800 ms for long-stop). We used $\omega_p = 1.5$, $\omega_v = 1$ and $\omega_f = 0.1$ as the values for the state cost parameters in this model, and the activation cost $R = 3 \times 10^{-6}$. Furthermore, in order to better match the forward velocity profiles, we also introduced a non-stationarity in the activation cost $R$ of the long stop movement, where the total integral of the activation cost over the movement is not changed, but this cost develops over time during the movement. Specifically, at a time $t$ in the trial, the activation cost for the long-stop movement was computed by:

$$R_{long-stop}(t) = RC(t), \tag{4}$$

where

$$C(t) \propto \exp\left(p\frac{t+q}{r}\right), \tag{5}$$

and the mean of $C(t)$ equals 1 for the duration of the trial, so that $R_{long-stop}$ produces the same amount of activation as $R$ over the duration of the trial. Here $p = 1$, $q = $ -1000 and $r = 65$ are constants, fit via trial and error in order to produce the forward velocity profile of long-stop condition that matches the velocity of stop and hit conditions. We have previously shown that such modulation only affects the kinematics of the movement, but does not change the feedback responses when expressed against the time-to-target [38]. On the other hand, in order to incentivise the hit controller to produce faster movements at the target, we reduced the cost parameters for terminal velocity and terminal force by a factor of 50. As a result, such controller produced hit-like movements that were aimed directly at a target, positioned at 25 cm distance, over 620 ms, which matched the kinematics of the long-stop controller over this movement segment.

Finally, for each controller we simulated feedback response intensity profiles along the movement, which we then used to compare the control policies predicted by each controller. To do so, we induced lateral target perturbations of 2 cm magnitude during the simulated movement to the target and recorded the corrective force, produced by each controller as a result of these perturbations. While in the experimental study we only induced such perturbations at five different onsets due to practical reasons, in our simulations we could perturb the movements at every point in time and fully map the response intensity profiles over the movement. Thus, for each model we simulated different movements with perturbations at each movement time-step (i.e. every 10 ms), with one perturbation only happening once per movement. In addition, to simulate the visuomotor delay that is present in humans, we delayed the onset of each perturbation by 150 ms, so that for the perturbation triggered at time $t$, the target is shifted at time $t + 150$ ms. We then averaged the force, produced by our model over a time interval 10 ms–60 ms after the target was shifted (160 ms–210 ms after the perturbation was triggered), representing the visuomotor response window of 180 ms–230 ms in human subjects. Note that we used an earlier window for the model simulations than for the human subjects as the responses in the simulations ramp up fast due to muscles simplified to a single low pass filter.

## Supporting information

**S1 Text. Initial learning of feedback controllers.** Here we verify whether our participants developed different feedback controllers for hit and stop tasks over the blocked schedule, or if these controllers were innate. To do so, we analyse the visuomotor responses in the first few blocks of the study, showing that these responses can be considered innate.
(PDF)

**S2 Text. Effect of condition clustering in mixed schedule.** Visuomotor responses were analysed in trials immediately following the condition switch (hit to stop or stop to hit) in mixed schedule. Analysis shows same regulation as in the entirety of the mixed schedule, implying rapid switching.
(PDF)

**S1 Fig. OFC model simulations in channel trials.** Model simulations performed in cbannel trials, instead of free movement responses. **A**. Velocity profiles for stop (blue), hit (red) and long-stop (green) conditions. **B**. Model simulations of feedback intensities as a function of time-to-target and **C**. position for the three conditions. Simulations in the channel trials qualitatively predict the same regulation as do the simulations of free movements.
(EPS)

## Acknowledgments

We thank Hanna Hoogen, Isabelle Hoxha and Oliver Gerke for contributions to preliminary projects related to this manuscript. We thank Clara Günter, Jing Zhang, Sae Franklin, and Marion Forano for their feedback on this manuscript.

## Author Contributions

**Conceptualization:** Justinas Česonis, David W. Franklin.

**Data curation:** Justinas Česonis.

**Formal analysis:** Justinas Česonis.

**Investigation:** Justinas Česonis, David W. Franklin.

**Methodology:** Justinas Česonis, David W. Franklin.

**Project administration:** Justinas Česonis.

**Resources:** David W. Franklin.

**Supervision:** David W. Franklin.

**Validation:** Justinas Česonis, David W. Franklin.

**Visualization:** Justinas Česonis.

**Writing – original draft:** Justinas Česonis.

**Writing – review & editing:** Justinas Česonis, David W. Franklin.

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
