## [Decision Letter · Decision Letter 0]

2 Feb 2022

Dear Mr Česonis,

Thank you very much for submitting your manuscript "Contextual cues are not unique for motor learning: Task-dependant switching of feedback controllers" for consideration at PLOS Computational Biology.

As with all papers reviewed by the journal, your manuscript was reviewed by members of the editorial board and by several independent reviewers. In light of the reviews (below this email), we would like to invite the resubmission of a significantly-revised version that takes into account the reviewers' comments.

Overall, the reviewers were generally positive about the paper and found the paper to be technically sound. Reviewer 2, however, raised an important issue about the novelty of the work, pointing out several papers that have already provided evidence for the general claim that people are able to flexibly switch between different controllers in a task-dependent manner. I believe (and the reviewer seems to agree) there is sufficient novelty in the current paper to warrant publication. However, I agree with Reviewer 2 that the paper needs to better acknowledge previous studies that have already addressed the general question of switching between feedback controllers (and reached the same conclusion), and should more clearly distinguish the specific contribution of this paper in relation to that prior work.

We cannot make any decision about publication until we have seen the revised manuscript and your response to the reviewers' comments. Your revised manuscript is also likely to be sent to reviewers for further evaluation.

Sincerely,

Adrian M Haith

Associate Editor

PLOS Computational Biology

Samuel Gershman

Deputy Editor

PLOS Computational Biology

Reviewer's Responses to Questions

**Comments to the Authors:**

Reviewer #1: This study has demonstrated that the motor system can flexibly recruit different feedback controllers depending on the type of task (hitting and stop tasks). It seems that previous studies have already demonstrated this. However, the authors pointed out that this was not true: The identical feedback controller could produce different feedback responses, and therefore observing different feedback responses is not solid evidence for the recruitment of different feedback controllers. Interestingly, their previous study (Cesonis and Franklin 2020) had demonstrated that the identical feedback controller induced the identical feedback responses when they were examined at the same time-to-target.

They experimentally demonstrated that different feedback responses were induced for the hitting and reaching tasks, even when the responses were evaluated at the same time-to-target and that this result was reproduced by the OFC models with different cost functions for both tasks. Furthermore, the schedule of tasks (i.e., block or mixed schedule) did not influence the result, enabling them to conclude that the motor system can flexibly and rapidly switch between different feedback control policies.

It is well-known that the motor system can create and retrieve different feedforward controllers when the appropriate cue is provided. This study is a significant contribution demonstrating that the motor system also has the ability to switch between different control policies depending on conditions. I was also intrigued by the idea that the feedback response at the same time-to-target can be used as a probe that examined the feedback controllers. Since this study was well-designed, I only have a few very minor issues.

1. Long-stop condition: This is an important control condition. The simulation result was presented, but the experiment was not performed in this condition. Do the authors expect that the experimental result would match the simulation?

2. Effect of schedule: I understood that there were no systematic differences between the blocked and mixed schedule (Fig.4). I do not oppose the idea of rapid switching, but I am curious about if the aftereffect from one condition to another did not exist. In the mixed schedule in which the hitting and stop tasks were randomly changed, one condition should continue several trials, which might obscure the effect of interference. The difference could be observed when analyzing the data in the trials only after switching?

3. Fig.2.B, C and 3E, F: The intensity was likely to be calculated as an averaged feedback response (i.e., force) in 180ms and 230ms from the perturbation onset. There seemed some inconsistencies between Fig.2 and 3. The force output shown in Fig.2 was approximately 0.3N for the earliest perturbation, while the intensity in Fig.3 was 0.8N. Furthermore, the difference in the force output between both tasks (Fig.2B and C) is much greater than the difference in the intensity (Fig.3E, F).

4. line 117-118 on page 9: There are garbled characters before 1cm/s and 20cm/s.

5. Line 356 on page 12: “In a recent study” should be “A recent study”?

Reviewer #2: The paper by Cesonis and Franklin presents data about changes in control policies between trials that differed based on their instructions. In one set of trials, participants were instructed to reach and stop at a target. In another set of trials, they were instructed to hit the target. Based on the premise that these different instructions translate into flexible changes in control strategies, it is shown that participants indeed switched controller from one trial to the next. The paper is interesting and all technical aspects seem to have been performed well. But it its current form I cannot endorse the paper because it ignores or dismisses previous work on the same topic.

As is, the paper cannot be accepted because it is not novel. A first point is the switching of controllers across trials, which has long been known. In fact, it has been included in the experimental design of many previous studies. Without putting emphasis on it, Nashed and colleagues (2012) already analysed the difference between blocked and random design to highlight trial-by-trial changes in goal structure. Later, Lowrey and colleagues (J Neurophysiol, 117: 1070–1083, 2017) used a random design with trials towards a round target or a bar target randomly interleaved and highlighted task-dependent switching of controllers. Orban de Xivry (PloS one 8 (6), e66013) compared switching of controllers with a blocked or random schedule and found a switch albeit smaller. The switching per se is not new.

More striking was the effort to minimize our recent study investigating switching of controllers across and within trials by de Comité et al. (125 (5), 1883-1898), which again showed the same results. I find that the authors dismissed our previous findings. First, we included also a change in controller across trials in the absence of online perturbation of the target shape as a control condition, reproducing the results of Nashed, Lowrey and colleagues. Then, it is written “the change in control signals alone does not directly imply a change in the feedback controller”. Yet, in this previous study, the change in control was evoked by a change in the structure, not the state (for instance position and velocity) of the goal. The authors then argue vaguely that this change in control can result from a change in the state that includes the target representation. The statement is inaccurate at best, or misleading if intentional. I believe the authors have missed that, computationally, changing the structure of the target from a square to a bar corresponded to a change in task demand. The critique was unfair and the author did not give proper credit to this previous work.

Towards assessing the paper as objectively as possible, it must be recognized that computationally again, our previous experiment corresponded to manipulating the factor called wp,t in their equation (3), which either has two non-zero components (penalizing x, and y) or one (penalizing only y). What the authors have done is testing the same idea by manipulating the other factors weighting velocity and force in the cost function. It is an important and interesting contribution on its own, but it does test the same idea only with other state variables. As such it deserves publication provided that previous contributions are properly acknowledged. The change of controllers has been demonstrated before, this is not new, what is tested here is whether this result also applies to hit and stop conditions, which are believed to require a change in the controller not linked to target geometry but to the penalty on the final state. Both manipulations imply a change in cost-function that must translate into an update of feedback gains. A substantial reworking is necessary to make this aspect very clear and upfront.

Specific points:

I did not find any definition of the time-to-target in the data, apologies if I missed it. Since there is no variability in time-to-target of Fig. 4b, I am assuming that it is measured as the difference between the perturbation time and some time limit, but since the perturbation were triggered in position I failed to understand how there was no variability in this quantity.

The authors equate urgency and time-to-target, which is a bit hazardous because the urgency may refer to an instantaneous pressure signal, without predicting how long or how far in time will movements be completed.

With respect to computational modelling I had concerns about the comparison made between the forces measured experimentally in a force channel and the forces produced in simulations. Was there a simulation of a force channel? Is there an impact on the correspondence between measured and simulated variables? Pls. add justification to the paper.

I found that the overall narrative was sometimes unclear. It has been known for a long time that between blocks of trials with different instructions, humans can change their goal-directed policy. This in my view is the main contribution of the use of optimal feedback control. In contrast, in the literature on motor adaptation shows gradual changes in internal models in ABA learning designs, with unlearning and interferences. Thus, it seemed clear that internal models for reaching do not change as quickly as task-dependent policies. The case of motor adaptation requires special contextual cues that seem less important for task-related constraints. The analogy is also limited, as context-dependent motor adaptation is associated with different feedforward controllers, while in the case of change in task-demands the authors use a model without feedforward pass. If the results must be interpreted in light of context-dependent learning I admit I see more differences between these paradigms than what is suggested.

Respectfully submitted,

F. Crevecoeur

Reviewer #3: In this paper the authors investigate to what extent healthy participants can switch between feedback controllers across tasks. This issue is clearly of interest to the motor control community. Indeed, although previous work has shown that task-dependant switching of feedforward controllers was possible, to my knowledge this issue has never been addressed in the context of feedback controllers. By comparing how participants react to visual perturbations (i.e. unexpected lateral target jumps) when performing either a reaching (stop) or an intercepting (hit) task, the authors can probe that these two tasks rely on separate feedback controllers. Importantly, evidence for separate controllers can be found even when participants are required to alternate reaching and intercepting trials. The authors conclude that contextual switching of feedback controllers is possible (as previously observed for feedforward controllers). In general, I found the manuscript very well written, with clear objectives, using a careful/clever design, and offering a lot of technical details. Still, I believe that this manuscript can be improved in several places (see my comments below).

Major Comments

1) Methodology (line 593-596): The authors state that all participants (n=14) performed first the blocked-design conditions then performed the mixed-design conditions (alternation of tasks). I wonder to what extent this design has favored the emergence of separate feedback controllers. When a participant performs a block of 416 trials while executing the same task over and over, it seems adequate to develop a feedback controller that is specific to this task. When the participant is subsequently asked to alternate trials in one task with trials in the second task, one may assume that those preexisting (separate) controllers may facilitate the switching of controllers across tasks. Now it is less clear what would have happened if participants had been exposed first to the mixed-design schedule (rapid alternation of tasks). One may wonder whether the emergence of separate controllers would have been possible. Although, I feel confident with the observations reported by the authors, I believe this extra information would impact the extent of the conclusions formulated by the authors. Obviously, I understand that investigating this issue necessitate to collect extra participants (but one may argue that the current number of participants is on the low side), but if such material is not provided, the authors should at least acknowledge the limitations of their experimental design.

2) Data analysis: In Fig 4B feedback response intensities are provided in each task (hit and stop) and for each schedule (blocked vs. mixed). Visual inspection suggests that the difference across tasks is greater in the blocked design (red vs. dark blue curve) than in the mixed design (orange vs. light blue). I wonder if the authors performed some statistical analyses that addressed specifically this issue (apart from the lack of condition/schedule interaction), as a smaller disparity across controllers matches with one the hypotheses formulated initially. One would also appreciate further analyses of the mixed schedule to assess the presence of possible carry-over effects when switching across tasks. For instance, one could assess whether the gain of feedback responses in a stop/reach trial increases as the number of preceding hit/intercepting trials increases. A possible lack of carry-over effects across tasks would strengthen the conclusions made by the authors (i.e. fully independent feedback controllers allowing instantaneous switch).

3) Learning: As far I understand, all available (perturbed) trials were considered to assess the visuofeedback gains associated with each task. It is unclear to what extent these responses are modulated by prior experience with the task (and perturbation). It would be helpful to clarify whether these feedback controllers were preexisting/innate, or whether they slowly build up during practice with the task. One could imagine that during initial training, feedback controllers were rather similar for both tasks, but progressively diverged with further experience. Providing some information about the time course of feedback responses over the blocked schedule would be helpful. For instance, one could contrast responses in early and late trials. Obviously, I understand that the statistical power will drop, but clarifying how these feedback controllers appeared/emerged in the first placed seems critical.

Minor comments

Line 117-118: Remove the extra characters preceding the value of velocity

Line 330: Please consider that not all studies support the notion that feedback and feedforward mechanisms are necessarily shared (Abdelghani and Tweed 2010; Gritsenko and Kalaska 2010; Kasuga et al. 2015; Yousif and Diedrichsen 2012).

Line 418: The authors emphasized that “time-to-target” is a better reference frame to monitor feedback responses. This variable resonates with another variable “time-to-contact” (Tau) that is very popular in the field of ecological psychology. I am wondering if the authors considered a possible connection with this field, or this is just a fortuitous coincidence?

Line 505: Was there a rational for the number of participants (n=14)? Nowadays, more and more studies are considering sample size of at least 20 participants.

Line 536: In the intercepting task, people were given feedback to hit the target at a given speed, but were they given explicit instructions regarding the fact that their peak velocity had to coincide or not with the hit?

Line 598: Did you have any constraints regarding the randomization of task order? For instance, was it possible for participants to experience 3 (or possibly more) consecutive trials of the same task? Did you set a limit for this (i.e. maximum number of consecutive trials in the same task)? More information on this issue would be helpful in the method.

Line 631: I assume that the neutral force profile was obtained from the non-perturbed trials performed with the channel, correct? I presume this force profile was non monotonic (possibly velocity dependent), correct? How did you deal with the fact that movement time varied across trials?

Line 637: Although I do understand the rational for the normalization of force responses across participants, it does not seem to match with the unit employed in Figure 3 and 4. In both figures the intensity of feedback responses is presented in Newtons. Please clarify how the normalization procedure allows the preservation of feedback responses in Newtons.

Fig 1: The authors contrast two extreme situations: a single versus 2 independent controllers. One could imagine an intermediate scenario in which the single controller carries history dependent effect (see also my second main comment on this). Perhaps the authors could acknowledge this third option in the caption or somewhere in the text.

Fig 3A&D: For both tasks, peak velocity is about 10% smaller in the experimental data than in the model predictions (60 cm/s). Even if the predictions would probably be qualitatively similar, I wonder why the authors did not readjust the settings for the model predictions.

Figure 3B: It is a bit odd to provide a single label (B) to account for two graphs (time-to-target and time-to-movement end) as opposed to all the other graphs (one label per graph). I would suggest inserting an extra label for the neighboring graph (with final label reaching G instead of F).

Fig 3BC: The model predictions suggest that sometimes the intensity of feedback responses can be negative (below 0). This feature is not visible in the experimental data (see Fig 3E) which presumably makes sense given the way the force data were analyzed (as force direction did not seem to matter). Altogether I am left wondering how to reconcile these separate observations. Please clarify.

Fig 3E+Line 153: This graph clearly shows different feedback responses across tasks, and the authors capitalize on these differences. Apart from being consistent with optimal feedback control, can the authors provide a (simple) rational as to why, during the first part of the movement, feedback responses should be larger for the hit condition than for the stop one?

**Have the authors made all data and (if applicable) computational code underlying the findings in their manuscript fully available?**

Reviewer #1: Yes

Reviewer #2: Yes

Reviewer #3: Yes

PLOS authors have the option to publish the peer review history of their article (what does this mean?). If published, this will include your full peer review and any attached files.

Reviewer #1: No

Reviewer #2: No

Reviewer #3: No
---

## [Decision Letter · Decision Letter 1]

5 Apr 2022

Dear Mr Česonis,

Thank you very much for submitting your manuscript "Contextual cues are not unique for motor learning: Task-dependant switching of feedback controllers" for consideration at PLOS Computational Biology. As with all papers reviewed by the journal, your manuscript was reviewed by members of the editorial board and by several independent reviewers. The reviewers appreciated the attention to an important topic. Based on the reviews, we are likely to accept this manuscript for publication, providing that you modify the manuscript according to the review recommendations.

The reviewers were generally very satisfied with the revisions to the manuscript. Reviewer 2 raised some remaining concerns, however. I don't think these are major issues, but I hope you might be able to address them (as well as those raised by Reviewer 3) by providing a response to the reviewers' concerns and revisions to the manuscript where these may be appropriate.

Sincerely,

Adrian M Haith

Associate Editor

PLOS Computational Biology

Samuel Gershman

Deputy Editor

PLOS Computational Biology

[LINK]

Reviewer's Responses to Questions

**Comments to the Authors:**

Reviewer #1: I was satisfied with the revision. I have no further additional comments.

Reviewer #2: The authors have clarified their contribution in regard of previous work investigating changes in target structure and the paper has improved. There remain points of disagreement that I would recommend the authors to consider.

Regarding the impact of the channel I am surprised to see the arguments developed, visuomotor delays are closer to 100ms than 150ms, see for instance Izawa and Shadmehr (JNS, 2008) who documented the effect of a jump on acceleration. Regarding the stretch response, the statement seems inaccurate in my view because the smallest perturbation loads applied very early during reaching are known to evoke short and long-latency responses (J Neurophysiol, 2012, 107: 2821-2832). In all it seems that there can be a feedback-mediated response to the channel, which is not captured in the simulation since there was no difference with or without channel. I believe that this discrepancy between model and data could be expanded a bit. It is strange to argue that the experiments aimed at probing control while using channels, then use a model without channel to simulate control and verify that results in the channel are not different.

I did not follow the authors’ response regarding the difference between change in control and adaptation, and I did not see any addition to the manuscript about this point. An indirect argument is made about the fact that the task was not novel but to my knowledge, dual adaptation can be compromised during exposures longer than the training sessions used here. In addition, the authors suggest that it would be an interesting control to test first the random schedule. Indeed, and I would be willing to bet that it works just fine or to a slightly lesser extent. On the contrary dual adaptation builds up over lover periods of time, and visual cues (and others) have been shown ineffective in evoking different contexts or goals. These are clear differences and I suggest to discuss the limitations of the analogy with context-dependent adaptation in more detail.

Reviewer #3: I am generally satisfied with the responses/changes provided by the authors to my concerns. Specifically, I appreciated their effort to assess the presence of possible carry-over effects across tasks. Evidence for instantaneous switching is strengthened. I have only a few minor suggestions left to improve their manuscript.

1) Fig S1: Please state the nature of error bars (SEM or 95% confidence interval)

2) Fig S2: It I unclear why data on panel B is displayed in arbitrary unit, whereas it is displayed in raw unit (N) in panel C. The resolution of the Y axis in panel C is low compared to other panels (2 versus 6 values). Please accommodate graph C accordingly. As evoked previously for Fig S1, please state the nature of error bars (SEM or 95% confidence interval).

**Have the authors made all data and (if applicable) computational code underlying the findings in their manuscript fully available?**

Reviewer #1: Yes

Reviewer #2: Yes

Reviewer #3: Yes

PLOS authors have the option to publish the peer review history of their article (what does this mean?). If published, this will include your full peer review and any attached files.

Reviewer #1: No

Reviewer #2: No

Reviewer #3: **Yes: **Frederic Danion

Figure Files:

Data Requirements:

Reproducibility:

References:

---

## [Editor Report · Decision Letter 2]

9 May 2022

Dear Mr Česonis,

We are pleased to inform you that your manuscript 'Contextual cues are not unique for motor learning: Task-dependant switching of feedback controllers' has been provisionally accepted for publication in PLOS Computational Biology.

Best regards,

Adrian M Haith

Associate Editor

PLOS Computational Biology

Samuel Gershman

Deputy Editor

PLOS Computational Biology

---

## [Editor Report · Acceptance letter]

6 Jun 2022

PCOMPBIOL-D-22-00004R2 

Contextual cues are not unique for motor learning: Task-dependant switching of feedback controllers

Dear Dr Česonis,

I am pleased to inform you that your manuscript has been formally accepted for publication in PLOS Computational Biology. Your manuscript is now with our production department and you will be notified of the publication date in due course.

With kind regards,

Olena Szabo
